# Predicting the Metastatic Potential of Papillary Thyroid Microcarcinoma Based on the Molecular Profile of Preoperative Cytology Specimens

**DOI:** 10.3390/ijms26136418

**Published:** 2025-07-03

**Authors:** Sergei A. Lukyanov, Sergei E. Titov, Aria V. Dzodzaeva, Vladimir E. Vanushko, Dmitry G. Beltsevich, Yuliya A. Veryaskina, Semyon V. Kupriyanov, Ekaterina V. Bondarenko, Ekaterina A. Troshina, Liliya S. Urusova, Sergei V. Sergiyko

**Affiliations:** 1Department of General Surgery, South Ural State Medical University, 454141 Chelyabinsk, Russia; ssv_1964@mail.ru; 2Department of the Structure and Function of Chromosomes, Institute of Molecular and Cellular Biology, Siberian Branch of the Russian Academy of Sciences, 630090 Novosibirsk, Russia; titovse78@gmail.com (S.E.T.); microrna@inbox.ru (Y.A.V.); 3AO Vector-Best, 630117 Novosibirsk, Russia; 4National Medical Research Center for Endocrinology, 115478 Moscow, Russia; ariyadz@mail.ru (A.V.D.); vanushko.vladimir@endocrincentr.ru (V.E.V.); belcevich.dmitry@endocrincentr.ru (D.G.B.); bondarenko.ekaterina@endocrincentr.ru (E.V.B.); troshina.ekaterina@endocrincentr.ru (E.A.T.); urusova.liliya@endocrincentr.ru (L.S.U.); 5Laboratory of Gene Engineering, Institute of Cytology and Genetics, Siberian Branch of the Russian Academy of Sciences, 630090 Novosibirsk, Russia; 6Laboratory of Evolutionary Cytogenetics, National Research Tomsk State University, 634050 Tomsk, Russia; pfft@mail.ru; 7Institute of Clinical Morphology and Digital Pathology, I.M. Sechenov First Moscow State Medical University, 119991 Moscow, Russia

**Keywords:** metastatic, papillary thyroid microcarcinoma, molecular testing, *HMGA2*, *TIMP1*, *FN1*, miRNA-146b, miRNA-7, miRNA-148b, *SLC26A7*

## Abstract

The strategy of active surveillance for papillary thyroid microcarcinoma (PTMC) is becoming increasingly popular within the global medical community. A key criterion for selecting this strategy is the absence of any signs of lymphogenic or distant metastases. The present study assessed the diagnostic accuracy of molecular genetic markers for predicting the metastatic potential of patients with PTMC. We evaluated the expression levels of 33 molecular genetic markers in cytology samples from 92 patients with PTMC and confirmed histological diagnosis. Among these patients, 32 had metastases to regional cervical lymph nodes. Our findings revealed the upregulated expression of the *HMGA2*, *TIMP1*, and *FN1* genes, as well as microRNA-146b, in patients with metastatic PTMC. Conversely, we found the downregulated expression of miRNA-7 and -148b in metastatic tumors. In metastatic tumors, significant reductions were observed in *DIO1* activity (11-fold), *TFF3* gene expression (8-fold), *TPO* expression (4-fold), and *SLC26A7* expression (2.6-fold). All the markers exhibited high sensitivity (84.5–90.6%) in detecting metastatic PTMC, although the specificity proved to be low. The use of molecular markers to predict lymphogenic metastatic spread in patients with PTMC could enhance existing risk grading systems. Such assessments can already be applicable at the preoperative stage.

## 1. Introduction

Papillary thyroid microcarcinoma (PTMC) is among the most frequently diagnosed forms of thyroid cancer, with recent decades witnessing an upward trend in incidence. From 1990 to 2017, the global annual age-standardized incidence rate (ASIR) of thyroid cancer increased from 2.1 to 3.15, with an annual growth rate of 1.59% [1]. The increasing number of diagnosed cases can be attributed to several factors, including advancements in diagnostic procedures and a rise in the frequency of fine-needle aspiration biopsy (FNAB) utilization. Contemporary imaging techniques and diagnostic approaches have enabled the detection of even minute lesions in the thyroid gland, consequently leading to an increase in the number of diagnosed cases of PTMC [2].

There is a tendency for specialists to perform FNAB of even minor thyroid lesions despite the absence of consensus in clinical guidelines regarding the necessity of such procedures [3]. Thus, given even the slightest suspicion of malignancy, biopsies are conducted, thus leading to a rise in the number of diagnosed PTMC cases. Surgical interventions for thyroid conditions are frequently guided by FNAB results, thus contributing to an increased number of such procedures [4]. There is a controversy among expert opinion on this approach due to evidence indicating that a significant number of detected microcarcinomas re-main stable and do not progress over extended periods [2]. Currently, the strategy of active surveillance (AS) for PTMC is gaining increasing popularity in the global medical community [5].

Research has shown that only 5% of all papillary microcarcinomas progress and necessitate active interventions [6]. However, this seemingly small percentage remains clinically significant, as tumor progression poses a considerable threat to patient well-being. The successful implementation of the AS strategy, therefore, requires close monitoring and regular evaluation to allow for the prompt detection of any changes in the tumor and facilitate timely intervention. Active surveillance of patients is currently predicated on tumors with the following characteristics: a diameter of less than 1 cm and a location distant from the trachea, esophagus, and laryngeal nerves [7].

Of particular importance is the patient’s psychological readiness for active surveillance. It is imperative that patients receive comprehensive information regarding the potential benefits and risks associated with this approach, along with the requirement for routine monitoring and examinations [8].

Notwithstanding the significance of the aforementioned criteria for choosing a strategy for active surveillance of papillary thyroid microcarcinoma, it must be acknowledged that these criteria are relative and subjective. Every instance of papillary thyroid microcarcinoma exhibits unique characteristics, and its progression can be affected by multiple factors [9].

The selection of an active surveillance strategy is contingent upon the absence of any indications of lymphogenic or distant metastases. In the event of metastases emerging during AS, a more aggressive treatment plan, including thyroidectomy, lymphadenectomy, and radioiodine therapy, becomes imperative. This treatment plan is more intensive than the one that could be used for a newly diagnosed microcarcinoma, with hemithyroidectomy typically a sufficient treatment option [4]. The quest for novel markers with which to predict the aggressiveness of PTMC as early as the preoperative stage remains an urgent and unresolved problem. Several studies [10,11] have investigated the prognostication of papillary thyroid cancer (PTC) aggressiveness through molecular genetic analysis. These studies have led to the establishment of molecular risk groups (MRGs). Specifically, the low-risk MRG exhibited *RAS*-like alterations, while the intermediate-risk MRG demonstrated *BRAF* alterations [12]. The high-risk profile was characterized by the presence of mutations in the *TERT*, *TP53*, *AKT1*, and *PIK3CA* genes. The molecular profile was shown to predict the risk of distant metastases with a reasonable degree of accuracy based on the presence of *TERT*, *TP53*, *AKT1*, and *PIK3CA* mutations [13,14]. This increase in accuracy generally improves the estimation of recurrence risk, as per ATA guidelines [15]. Nevertheless, prognostication of cervical lymph node metastasis based solely on mutational data proved unfeasible.

The associations that have been revealed between the presence of regional metastases in patients with PTC and the miRNA + mRNA expression levels in tumor tissue are of particular interest. However, the feasibility of detecting these molecular genetic alterations in FNAB specimens has not been assessed in these studies [16,17,18,19].

Our previous study investigated the potential of various molecular markers (e.g., miRNA and protein-coding gene expression) for preoperative prediction of lymphogenic metastasis and extrathyroidal invasion in PTC [20]. The present investigation assesses the diagnostic accuracy of molecular genetic markers in predicting the metastatic potential of patients with PTMC.

## 2. Results

The study population included 92 patients, with 81.5% being female and 18.5% being male. The mean age of patients undergoing surgery was 47 years (range: 18–76 years). The prevalence of PTMC was found to be most significant in the right thyroid lobe, with a 70% occurrence, while the left thyroid lobe exhibited a 30% prevalence. According to the standardized European Thyroid Imaging Reporting and Data System (EU-TIRADS), six (6.5%) patients were classified as EU-TIRADS category 3, 19 (20.7%) patients were classified as EU-TIRADS category 4, and 67 (72.8%) patients were classified as EU-TIRADS category 5. According to the Bethesda System for Reporting Thyroid Cytopathology (TBSRTC, 2023), the distribution of the analyzed patient group was as follows. Atypia of Undetermined Significance (AUS) was identified in 2 (2.2%) patients, Follicular neoplasm (FN) was diagnosed in 13 (14.1%) patients, Suspicious for Malignancy (SM) was identified in 43 (46.7%) patients, and Malignant (M) was diagnosed in 34 (37%) patients. The surgical interventions performed included hemithyroidectomy in 60 (65.2%) cases, and thyroidectomy in 32 (34.8%) cases, with central lymphadenectomy conducted in 27 patients and a combination of central and lateral lymphadenectomy in 5 patients. The presence of metastases, as determined by ultrasound imaging or thyroglobulin measurement in the fluid removed during FNAB prior to surgery, served as an indication for lymphadenectomy. Prophylactic lymphadenectomy was not performed. The mean tumor nodule size was estimated to be 7 mm, with a range of 4–9 mm.

Statistically significant differences were detected for 11 out of 33 analyzed molecular genetic markers: miRNAs miRNA-146b, miRNA-7, and miRNA-148b, as well as messenger RNAs *FN1*, *TIMP1*, *TPO*, *SLC26A7*, *HMGA2*, *DIO1*, *TFF3*, and the *BRAF* mutation (Table 1).

In patients with metastatic PTMC, the expression levels of the *HMGA2* gene (a transcriptional regulation factor), the *TIMP1* gene (a matrix metalloproteinase inhibitor), and the *FN1* gene (encoding fibronectin) were found to be elevated, along with the upregulation of microRNA-146b. Downregulation of miRNA-7 and miRNA-148b was also detected in metastatic tumors, indicating their tumor suppressor roles. The presence of metastatic tumors was associated with a significant decrease in the activity of iodothyronine deiodinase (*DIO1*), with an average reduction of 11-fold, an 8-fold decrease in the expression of the *TFF3* gene, a 4-fold decrease in the expression of the thyroid peroxidase (*TPO*) gene, and a 2.6-fold decrease in the expression of the *SLC26A7* (the sulfate anion transporter) gene.

Given the allowance for multiple hypothesis testing with Bonferroni correction, the significance level *p* of 0.05/33 = 0.0015 was achieved exclusively for miRNA-146b (*p* = 0.013).

The *BRAFV600E* mutation was detected in 24 out of 32 patients with metastases and 34 out of 60 patients without metastases. The odds ratio, with a 95% confidence interval, was determined to be 4.6 (1.8–11.9), and the *p*-value was found to be 0.0016.

The areas under ROC curves (AUC) were calculated for all these 11 markers. This study found that the AUC values for miRNA-146b, miRNA-7, *HMGA2*, and miRNA-148b were greater than 0.7, indicating a satisfactory outcome (see Figure 1).

The optimal separation threshold for these markers was determined by calculating the Youden index. The diagnostic capability was assessed for each of these four markers and the *BRAF* mutation (Table 2). The presence of a detected *BRAFV600E* mutation, a miRNA-146b expression level ≥ 5.4, a miR-7 expression level ≤ 0.13, an *HGMA2* expression level ≥ 0.17, and a miRNA-148b expression level ≤ 0.67 was identified as a set of diagnostic factors for metastatic PTMC.

Except for the *BRAF* mutation, all markers demonstrated high sensitivity but low specificity (approximately 50%) in the detection of metastatic PTMC. In summary, the expression levels of miRNA and mRNA were consistent with metastatic carcinoma in most patients without metastases.

## 3. Discussion

The ATA recurrence risk stratification system assesses the likelihood of PTMC persistence to be low. Unifocal papillary microcarcinomas located inside the thyroid gland and exhibiting no aggressive histological signs are believed to be characterized by a recurrence risk of less than 1% [5]. According to the aforementioned guidelines, the recurrence risk is associated with the presence of cervical lymph node metastases. When assessing the recurrence risk, it is imperative to consider the number and size of metastases, the number of affected lymph nodes with extracapsular invasion, and their localization (in the central or lateral cervical region). Therefore, the histological variant and the stage of the neoplastic process can be identified based on the presence of capsular invasion, multifocality, and cervical lymph node metastases. However, only after the final histological examination can this determination be made. This issue constitutes a significant challenge for proponents of active surveillance in PTMC.

A comprehensive observational study of active surveillance for patients with PTMC was undertaken at the Kuma Clinic, following a suggestion from Akira Miyauchi [6]. In this study, 3222 patients with microcarcinomas were followed, and their outcomes were compared to those of 2424 patients who underwent surgical treatment. Over a 20-year period, the AS group demonstrated a statistically significantly increased incidence of cervical lymph node metastasis (1.7% vs. 0.7% in the group of operated patients). The authors, however, inferred a low likelihood of metastatic progression, thus deeming active surveillance appropriate. The literature suggests that the risk of metastatic dissemination is elevated in individuals under 30 years of age, of male sex, and with tumors measuring 6 mm or more in size [6,21].

The Memorial Sloan Kettering Cancer Center (MSKCC), in collaboration with the Kuma Hospital group, has developed a clinical protocol to evaluate the suitability of active surveillance for specific patients [7]. The “ideal” candidates for this procedure are patients over the age of 60 years who have a tumor that is not adjacent to the thyroid capsule. A “suitable” candidate is defined as a patient who meets specific criteria. These criteria include being younger and having a multifocal tumor with a potentially aggressive molecular phenotype, as indicated by the presence of *BRAF* and *TERT* mutations, as well as *PD-L1* positivity. The tumor must be located adjacent to the thyroid capsule, but it should not be situated within the risk zones. Additionally, the presence of other imaging features that could complicate further surveillance should be taken into consideration, such as thyroiditis or other benign thyroid nodules. An “ill-suited” candidate is defined as one with tumors in high-risk locations, such as near the trachea or the recurrent laryngeal nerve or with signs of extrathyroidal invasion. The authors of these recommendations note that many of these characteristics are rather relative and strongly dependent on the expertise of the clinic and its medical personnel. According to experts in the field, molecular genetic studies hold considerable promise as a means of identifying indications for using the active surveillance approach. However, these methods are not yet commonly used, especially in developing countries [22].

In the United States, cytological diagnosis of indeterminate thyroid nodules currently employs three commercially available molecular tests. Afirma GSC is an RNA-based genomic sequencing classifier that utilizes gene expression to ascertain the malignant potential of a node. ThyGeNext + ThyraMIR is a multiplatform test that encompasses the evaluation of genetic alterations in DNA, mRNA, and regulators of gene expression, known as microRNAs. Thyroseq GC is a multigenic genomic classifier. However, the limitations of these tests preclude their ability to provide insights into the risk of metastasis in papillary thyroid cancer [23]. We determined the molecular markers using real-time PCR. These studies are relatively inexpensive, straightforward to execute, and accessible in most PCR laboratories, including in developing countries. The aforementioned factors engender the possibility of conducting such testing in authentic clinical settings.

At present, *BRAF* mutation is the most extensively researched and readily accessible molecular genetic marker. A comprehensive meta-analysis [24] revealed a 43% increased risk of cervical lymph node metastasis in PTMC patients carrying the *BRAFV600E* mutation (OR = 1.43, 95% CI = 1.19–1.71). According to our data, the OR was 4.6 (CI = 1.8–11.9), which is probably related to the higher occurrence of this mutation (63%) in the study group. Should this marker serve as the sole criterion, 63% of PTMC patients would be a priori unsuitable for continued active surveillance.

A meta-analysis published in 2017 reported the potential role of microRNA-146b in oncogenesis of papillary thyroid cancer [25]. Its elevated expression level was associated with a more aggressive course of the disease. According to the findings of this study, the use of this marker for predicting metastatic spread has been shown to be more accurate than prediction based on *BRAF* mutation (87.5% sensitivity, 50% specificity). The remaining two microRNAs (miR-7 and miR-148b) and the *HMGA2* gene exhibited a comparable degree of accuracy. The high sensitivity of the test enables the accurate prediction of metastases in the neck lymph nodes in patients with papillary microcarcinomas. However, when applied universally to all patients eligible for active surveillance, the test will provide low specificity results, suggesting that approximately half of patients with PTMC should undergo surgery. Therefore, none of these indicators should be considered definitive and should only be used as supplementary methods in cases of uncertainty. For example, a case that meets these criteria might include a tumor with a size slightly exceeding 10 mm or the presence of cervical lymph nodes that are deemed to be suspicious according to ultrasound imaging.

Following the implementation of the Bonferroni correction, only miR-146b remained statistically significant (*p* < 0.0015). This finding suggests that some associations with other molecular genetic markers may be accidental. To test this hypothesis, larger-scale studies with a substantial number of patients will be required.

## 4. Materials and Methods

Between 2021 and 2023, 92 patients were chosen with PTMC, with 32 of them having metastases to regional cervical lymph nodes. The study design was approved by the Ethics Committee of the South Ural State Medical University on 18 April 2019 (Protocol No. 3). Each patient provided informed consent following a thorough explanation of all procedures. Archival cytological preparations obtained by FNAB were used to analyze the molecular profile. All the samples were examined by a morphologist and had sufficient cellular strength. Not requiring special storage conditions, stained cytological preparations were stored at room temperature. While the retrospective design of this study may have introduced some bias, its impact on the results is considered negligible.

The mRNA set was primarily selected based on the literature review. Protein-coding genes were chosen so that their exon–intron structure enabled the detection of mRNA without the preliminary removal of genomic DNA. The list of mRNAs included 18 genes: *FN1*, *GMNN*, *CDKN2A*, *TIMP1*, *CITED1*, *TPO*, *SLC26A7*, *HMGA*, *PD-L1*, *CPQ*, *RXRG*, *CLU*, *DIO1*, *TFF3*, *MEP*, *ECM1*, *SRPN*, and *TSHR*.

The microRNA set was selected based on our own data [26] and an analysis of existing literature data. A total of 13 microRNAs were included in experimental analysis: miR-146b-5p, miR-199b-5p, miR-221-3p, miR-223-3p, miR-31-5p, miR-375, miR-451a, miR-551b-3p, miR-185, miR-21, miR-148b-3p, miR-7m2, and miR-125b-5p. The BRAF V600E mutation and mtDNA were investigated independently. In total, 33 molecular genetic markers were analyzed in this investigation.

**RNA isolation.** The dried cytological preparation was washed into a tube containing three 200 μL portions of guanidine lysis buffer (4 M guanidine isothiocyanate, 25 mM sodium citrate, 0.3% sarkosyl, and 3% DTT aliquoted in an oxygen-free atmosphere, supplied by AO Vector-Best, Novosibirsk, Russia). The sample was thoroughly mixed and subsequently subjected to a thermal shaker for 15 min at a temperature of 65 °C. Subsequently, the tube was centrifuged at 10,000× *g* for 2 min. The supernatant was then transferred into new vials, followed by the addition of an equal volume of isopropanol. The reaction mixture was thoroughly mixed and left at room temperature for 5 min. After centrifugation for 10 min at 12,000× *g*, the supernatant was discarded, and the pellet was washed with 500 μL 70% ethanol and 300 μL acetone. Finally, the RNA was dissolved in 200 μL of deionized water. If not analyzed immediately, the RNA preparations were stored at −20 °C until further use.

**Semi-quantitative assessment of messenger RNA level.** All oligonucleotides were synthesized at AO Vector-Best (Novosibirsk, Russia). A semi-quantitative assessment of mRNA level was conducted using real-time RT-PCR with specific primers and fluorescent-labeled probes to detect the mRNAs of the respective genes and the phosphoglycerate kinase (*PGK1*) housekeeping gene, which was used as a normalization gene. The RT-PCR protocol was as follows: incubation at 45 °C for 30 min; heating at 95 °C for 2 min, followed by 50 cycles: denaturation at 94 °C for 10 s, annealing and elongation at 60 °C for 20 s. The relative expression level was calculated using the 2^−ΔCq^ method [27].

**Detection of microRNAs.** Real-time RT-PCR detection of 16 miRNAs was conducted for all types of tumors and lesions. Mature miRNAs were detected by stem-loop RT-PCR [28]. Real-time reverse-transcription PCR was conducted, as previously described [26]. A reverse transcription reaction followed by real-time PCR was performed individually for each miRNA. A single replicate of the analysis was performed for each sample. The miRNA level was normalized to the geometric mean of the levels of three reference miRNAs (miR-197-3p, -23a-3p, and -29b-3p) using the 2^−ΔCq^ method.

**Detection of somatic BRAF mutation.** All the samples were analyzed to detect somatic mutations, specifically V600E, V600K, and V600R, in the *BRAF* gene. Somatic mutations were detected by allele-specific PCR with a hydrolyzable probe. The PCR protocol was as follows: pre-heating at 95 °C for 2 min, followed by 50 cycles of denaturation at 94 °C for 10 s and annealing and elongation at 60 °C for 15 s.

A statistical data analysis was conducted using the SPSS Statistics 23 (IBM, Armonk, NY, USA) and Excel software (Microsoft, Redmond, WA, USA). The data are presented as the mean and median values, as well as the first (Q1) and third (Q3) quartiles. A comparative analysis of two independent groups for quantitative traits was performed using the Mann–Whitney U test. The multiple hypothesis testing problem was solved using the family-wise error rate (FWER) with Bonferroni correction. The significance level *p* was calculated as 0.05 divided by the number of features being compared. A total of 33 features were compared in our study. Therefore, the differences at significance level *p* < 0.05/33 = 0.0015 were considered statistically significant. The odds ratio (OR) with a 95% confidence interval (CI) was calculated to assess associations between two binary characteristics.

A ROC analysis was conducted to assess the predictive power of expression levels of various microRNAs and genes and to predict the risk of detecting lymphogenic metastases in PTMC. The operating characteristics of the following parameters were calculated: sensitivity (SEN), specificity (SPC), positive predictive value (PPV), and negative predictive value (NPV). Test comparison was conducted with allowance for the area under the ROC curves (AUC). The expert scale employed for the assessment of model performance was as follows. Scores ranging from 0.9 to 1.0 are considered to be of excellent caliber, while those falling between 0.8 and 0.9 are deemed to be very good. Scores from 0.7 to 0.8 are classified as good, with those ranging from 0.6 to 0.7 being considered fair. Scores falling below 0.6 are classified as unsatisfactory.

## 5. Conclusions

A growing trend among clinicians involves favoring AS over surgical intervention for patients presenting with PTMC. A primary concern in dynamic patient monitoring is the potential for cervical metastasis. Resection of a single thyroid lobe is deemed sufficient in cases of papillary microcarcinoma. Should cervical lymph node metastases be detected during surgery, total thyroidectomy and radioiodine therapy will be required. Molecular markers may offer supplementary prognostic value to existing risk stratification systems in predicting lymphogenic metastasis in patients with PTMC. These studies are straightforward to perform and readily accessible preoperatively. A predictive panel incorporating multiple markers may provide a future solution to this issue. The incorporation of mir-146 into prospective active surveillance studies is expected to enable a more accurate assessment of its prognostic value in the early development of cervical lymph node metastases in individuals with thyroid cancer.

## Figures and Tables

**Figure 1 ijms-26-06418-f001:**
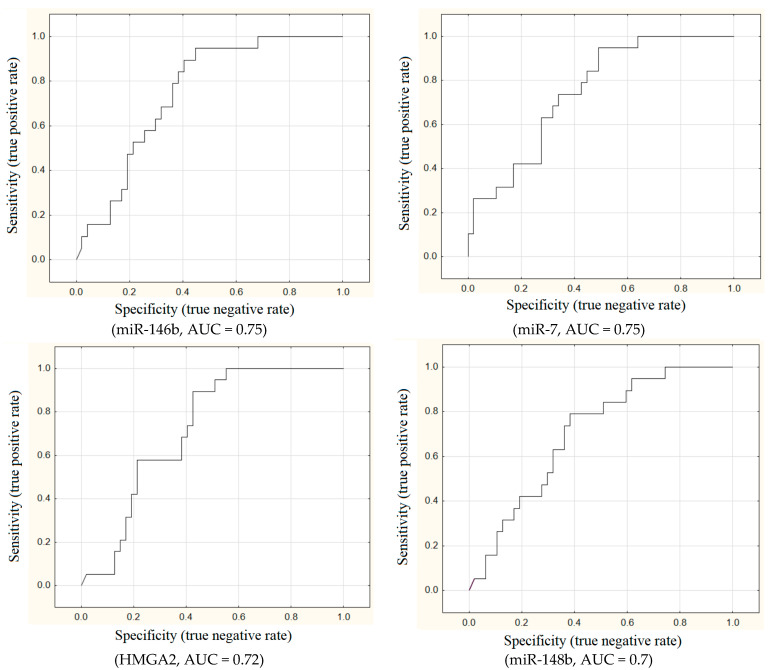
ROC analysis of molecular genetic markers of metastatic PTMC.

**Table 1 ijms-26-06418-t001:** Comparative analysis of patients with metastatic spread (n = 32) and without metastatic spread (n = 60) of PTMC.

Parameter	M, ME (Q1–Q3)		*p*, Mann–Whitney U Test
	N1a-b	N0	
miRNA-146b	10.6, 9.4 (7.4–13.1)	6.1, 5.5 (1.3–9.1)	0.0013
miRNA-7	0.03, 0.05 (0.03–0.1)	0.06, 0.12 (0.04–0.27)	0.0044
miRNA-148b	0.39, 0.42 (0.29–0.52)	0.7, 0.58 (0.38–0.96)	0.0052
*FN1*	80.6, 54.7 (32.6–110.7)	64.4, 27.6 (1.6–103.1)	0.049
*TIMP1*	5.1, 3.3 (2.3–6.1)	3.5, 1.9 (0.9–4.9)	0.013
*TPO*	1.9, 0.6 (0.2–3.7)	7.6, 1.3 (0.6–8.3)	0.026
*SLC26A7*	0.5, 0.3 (0.1–0.7)	1.3, 0.5 (0.2–1.4)	0.022
*HMGA2*	0.5, 0.4 (0.2–0.6)	0.3, 0.2 (0.1–0.4)	0.002
*DIO1*	0.01, 0.01 (0.002–0.02)	0.11, 0.01 (0.005–0.08)	0.018
*TFF3*	0.12, 0.04 (0.01–0.11)	1.0, 0.1 (0.03–0.3)	0.017

ME—median value; Q1, Q3—the first and third quartiles; M—mean value.

**Table 2 ijms-26-06418-t002:** Diagnostic characteristics of molecular genetic markers for detecting metastatic PTMC.

Parameter	Number of Observations; Parameter Value				
	miRNA-146b	miRNA-7	*HMGA2*	miRNA-148b	*BRAF*
FP	30	31	32	35	34
FN	4	4	3	5	8
TP	28	28	29	27	24
TN	30	29	28	25	26
SEN, %	87.5 (71–96.5)	87.5 (71–96.5)	90.6 (75–98)	84.5 (67.2–94.7)	75 (56.6–88.5)
SPC, %	50 (36.8–63.2)	48.3 (35.2–61.6)	46.7 (33.7–60)	41.6 (29–55)	43.3 (30.6–56.8)
PPV, %	8.4 (6.5–10.9)	8.2 (6.3–10.5)	8.2 (6.4–10.4)	7.1 (5.5–9)	6.5 (4.9–8.6)
NPV, %	98.7 (96.6–99.5)	98.7 (96.6–99.5)	98.9 (96.9–99.6)	98 (95.6–99.2)	97 (94.4–98.5)

FP—false positive; FN—false negative; TP—true positive; TN—true negative; SEN—sensitivity; SPC—specificity; PPV—positive predictive value; NPV—negative predictive value.

## Data Availability

Data is contained within the article.

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
