# Peer review of "Predicting the Metastatic Potential of Papillary Thyroid Microcarcinoma Based on the Molecular Profile of Preoperative Cytology Specimens"

_ijms, 2025, doi:10.3390/ijms26136418_

Round 1

Reviewer 1 Report

Comments and Suggestions for Authors

This study explores the potential of using molecular markers from preoperative fine-needle aspiration biopsy (FNAB) cytology samples to predict metastatic behavior in papillary thyroid microcarcinoma (PTMC). The authors analyzed 33 molecular genetic markers from 92 patient samples and found that certain mRNA and microRNA expression patterns, such as elevated HMGA2, FN1, TIMP1, and miR-146b, and reduced miR-7, miR-148b, DIO1, TFF3, TPO, and SLC26A7, correlate with the presence of lymph node metastases. The study suggests that these markers, especially miR-146b, may improve early risk stratification in PTMC patients.

The manuscript is well-organized and clearly written. The background is comprehensive and nicely sets the context. The methods are clearly described, and the results are well-supported by the data. However as per the iThenticate report the plagiarism index of 27% seems a bit towards higher end. Consider reducing the similarity index up to 20%. 

Major revisions

  1. While the sensitivity of most markers is quite high (~85–90%), their specificity is low (~40–50%). This could result in many false positives if applied clinically. The authors should emphasize this limitation more clearly in the discussion and caution against over-reliance on these markers alone.
  2. Only miR-146b remained statistically significant after Bonferroni correction (p < 0.0015). This suggests that some of the associations might be due to chance. The discussion should reflect on this and suggest validation in larger cohorts.
  3. Although molecular profiling seems promising, it remains to be seen whether it will be adopted in routine FNAB workflows. A brief paragraph on feasibility, cost, and accessibility of such testing in real-world clinical settings (especially in developing countries) would enhance the practical relevance.
  4. Consider validating these findings in a separate, independent cohort to strengthen conclusions.

Minor revisions

  1. A brief discussion on how these markers compare with other commercially available molecular panels (e.g., ThyroSeq, Afirma) would provide more context.
  2. Discuss whether combining multiple markers in a panel (rather than individually) might improve specificity.
  3. Ensure consistent formatting of gene and microRNA names (either miR or miRNA).
  4. Line 29: microNRA-7 should be corrected to microRNA-7.
  5. Line 82: The classification into “RAS-like,” “BRAF-like,” and “high-risk” is discussed broadly in literature. The authors might consider citing a relvant paper for classficiation here.
  6. Figure 1: The ROC curves are informative. It might help to add the AUC values directly to the figure for quick visual reference.
  7. Table 2: It would be useful to highlight which marker(s) passed Bonferroni correction here.
  8. Include confidence intervals for sensitivity and specificity in Table 2 to better express statistical reliability.
  9. Line 198–204: The suggestion to treat all patients with positive markers surgically may be premature. Authors could balance this with a discussion of integrating markers into broader clinical decision-making frameworks.
  10. Line 212: is it a heading ? if not then the first sentence needs to be rephrased.

Author Response

Response to Reviewer 1 Comments

1. Summary

We are grateful for the time and effort you have invested in reviewing this manuscript. The detailed responses can be found below, with the corresponding revisions/corrections highlighted/in track changes in the resubmitted files.

2. Questions for General Evaluation

Reviewer’s Evaluation

Response and Revisions

Does the introduction provide sufficient background and include all relevant references?

Can be improved

We have made additions to the introduction in accordance with your comments.

Is the research design appropriate?

Yes

Are the methods adequately described?

Yes

Are the results clearly presented?

Yes

Are the conclusions supported by the results?

Can be improved

We have made additions to the introduction in accordance with your comments.

Are all figures and tables clear and well-presented?

Yes

3. Point-by-point response to Comments and Suggestions for Authors

Comments 1: While the sensitivity of most markers is quite high (~85–90%), their specificity is low (~40–50%). This could result in many false positives if applied clinically. The authors should emphasize this limitation more clearly in the discussion and caution against over-reliance on these markers alone.

Response 1:      Thank you for pointing this out. We agree with this comment. Given that it should be considered in the discussion, we have expanded this part:

The high sensitivity of the test enables the accurate prediction of metastases in the neck lymph nodes in patients with papillary microcarcinomas. However, when ap-plied universally to all patients eligible for active surveillance, the test will provide low specificity results, suggesting that approximately half of patients with PTMC should undergo surgery. Therefore, none of these indicators should be considered definitive and should only be used as supplementary methods in cases of uncertainty. For example, a case that meets these criteria might include a tumor with a size slightly exceeding 10 mm or the presence of cervical lymph nodes that are deemed to be suspicious according to ultrasound imaging.

Comments 2: Only miR-146b remained statistically significant after Bonferroni correction (p < 0.0015). This suggests that some of the associations might be due to chance. The discussion should reflect on this and suggest validation in larger cohorts.

Response 2:

Thank you for pointing this out. We have added the necessary corrections to the discussion: Following the implementation of the Bonferroni correction, only miR-146b remained statistically significant (p < 0.0015). This finding suggests that some associations with other molecular genetic markers may be accidental. To test this hypothesis, larger-scale studies with a substantial number of patients will be required.

Comments 3:   Although molecular profiling seems promising, it remains to be seen whether it will be adopted in routine FNAB workflows. A brief paragraph on feasibility, cost, and accessibility of such testing in real-world clinical settings (especially in developing countries) would enhance the practical relevance.

Response 3:     

Thank you for pointing this out. We have added the necessary corrections to the discussion and conclusion:

We determined the molecular markers using real-time PCR. These studies are relatively inexpensive, straightforward to execute, and accessible in most PCR laboratories, including in developing countries. The aforementioned factors engender the possibility of conducting such testing in authentic clinical settings.

Comments 4: Consider validating these findings in a separate, independent cohort to strengthen conclusions.

Response 4:     

We totally agree with your suggestion and will definitely conduct future research on a larger sample. We have added the corrections to the conclusion.

Minor revisions

  1. A brief discussion on how these markers compare with other commercially available molecular panels (e.g., ThyroSeq, Afirma) would provide more context.

We have expanded the discussion:

In the United States, cytological diagnosis of indeterminate thyroid nodules currently employs three commercially available molecular tests. Afirma GSC is an RNA-based genomic sequencing classifier that utilizes gene expression to ascertain the malignant potential of a node. ThyGeNext + ThyraMIR is a multiplatform test that encompasses the evaluation of genetic alterations in DNA, mRNA, and regulators of gene expression, known as microRNAs. Thyroseq GC is a multigenic genomic classifier. However, the limitations of these tests preclude their ability to provide insights into the risk of metastasis in papillary thyroid cancer [26]. We determined the molecular markers using real-time PCR. These studies are relatively inexpensive, straightforward to execute, and accessible in most PCR laboratories, including in developing countries. The aforementioned factors engender the possibility of conducting such testing in authentic clinical settings.

  1. Discuss whether combining multiple markers in a panel (rather than individually) might improve specificity.

We have added:

A predictive panel incorporating multiple markers may provide a future solution to this issue.

  1. Ensure consistent formatting of gene and microRNA names (either miR or miRNA).

We have made the necessary changes and corrections.

  1. Line 29: microNRA-7 should be corrected to microRNA-7.

We have made the necessary changes and corrections.

  1. Line 82: The classification into “RAS-like,” “BRAF-like,” and “high-risk” is discussed broadly in literature. The authors might consider citing a relvant paper for classficiation here.

We have made the necessary changes and corrections.

  1. Figure 1: The ROC curves are informative. It might help to add the AUC values directly to the figure for quick visual reference.

We have made the necessary changes and corrections.

  1. Table 2: It would be useful to highlight which marker(s) passed Bonferroni correction here.

We have made the necessary changes and corrections.

  1. Include confidence intervals for sensitivity and specificity in Table 2 to better express statistical reliability.

We have made the necessary changes and corrections.

  1. Line 198–204: The suggestion to treat all patients with positive markers surgically may be premature. Authors could balance this with a discussion of integrating markers into broader clinical decision-making frameworks.

We have made the necessary changes and corrections.

  1. Line 212: is it a heading ? if not then the first sentence needs to be rephrased.

We have made the necessary changes and corrections.

Reviewer 2 Report

Comments and Suggestions for Authors

Tha authors studied 33 molecular genetic markers in cytological samples to predict the metastatic potential of papillary thyroid microcarcinoma by using PCR or RT-PCR techniques.

In clinical setting, it is difficlut to share the FNAB sample to perform molecular analysis since the FNAB may lack the cellularity. This is one reason why liquid biopsy is increasingly studied to supplement tissue biopsy to predict the prognosis. Furthmore, it is a retrospective study. For the above reasons, the originality of this article is banal.

In the section Materials and Methods, there is no information on how the cytology samples were conserved. The techniques of extraction of RNA or DNA were not described. For the miRNA level, the authors used the geometric mean of the levels of three reference miRNAs (miR-197-3p, -23a-3p, and -29b-3p) to normalize the miRNA level. It may be a novel method, but is this method valid ? One of most important questions is the quality of extracted RNA or DNA in cytological samples. The information on sample conservation is needed.

The conclusion is too general and not meaningful.

Author Response

Response to Reviewer 2 Comments

1. Summary

We are grateful for the time and consideration invested in reviewing this manuscript. The detailed responses can be found below, with the corresponding revisions/corrections highlighted/in track changes in the resubmitted files.

2. Questions for General Evaluation

Reviewer’s Evaluation

Response and Revisions

Does the introduction provide sufficient background and include all relevant references?

Yes

Is the research design appropriate?

Can be improved

We have made additions to the introduction in accordance with your comments.

Are the methods adequately described?

Must be improved

We have made additions to the introduction in accordance with your comments.

Are the results clearly presented?

Can be improved

We have made additions to the introduction in accordance with your comments.

Are the conclusions supported by the results?

Must be improved

We have made additions to the introduction in accordance with your comments.

Are all figures and tables clear and well-presented?

Yes

3. Point-by-point response to Comments and Suggestions for Authors

Comments 1: In clinical setting, it is difficlut to share the FNAB sample to perform molecular analysis since the FNAB may lack the cellularity. This is one reason why liquid biopsy is increasingly studied to supplement tissue biopsy to predict the prognosis. Furthmore, it is a retrospective study. For the above reasons, the originality of this article is banal.

Response 1:     

Thank you for pointing this out. We agree with this comment.

Between 2021 and 2023, 92 patients were chosen with PTMC, with 32 of them having metastases to regional cervical lymph nodes. The study design was approved by the Ethics Committee of the South Ural State Medical University on April 18, 2019 (Protocol No. 3). Each patient provided informed consent following a thorough explanation of all procedures. Archival cytological preparations obtained by FNAB were used to analyze the molecular profile..

Comments 2: In the section Materials and Methods, there is no information on how the cytology samples were conserved.

Response 2:

Thank you for highlighting this matter. We agree with this comment. We have added the necessary information:

Not requiring special storage conditions, stained cytological preparations were stored at room temperature.

Comments 3: The techniques of extraction of RNA or DNA were not described.

Response 3: Thank you for pointing this out. We agree with this comment. It is evident that the method in question was not initially described due to an oversight. An additional methodology has been incorporated into the Materials and Methods section.

Methods:

RNA isolation. The dried cytological preparation was washed into a tube containing three 200 μl portions of guanidine lysis buffer (4 M guanidine isothiocyanate, 25 mM sodium citrate, 0.3% sarkosyl, and 3% DTT aliquoted in an oxygen-free atmosphere, supplied by AO Vector-Best, Russia). The sample was thoroughly mixed and subsequently subjected to a thermal shaker for 15 minutes at a temperature of 65 °C. Subsequently, the tube was centrifuged at 10,000g for 2 minutes. The supernatant was then transferred into new vials, followed by the addition of an equal volume of isopropanol. The reaction mixture was thoroughly mixed and left at room temperature for 5 minutes. After centrifugation for 10 minutes at 12,000g, the supernatant was dis-carded, and the pellet was washed with 500 μL 70% ethanol and 300 μL acetone. Finally, the RNA was dissolved in 200 μL of deionized water. If not analyzed immediately, the RNA preparations were stored at -20 °C until further use.

Comments 4: For the miRNA level, the authors used the geometric mean of the levels of three reference miRNAs (miR-197-3p, -23a-3p, and -29b-3p) to normalize the miRNA level. It may be a novel method, but is this method valid?

Response 4:

Thank you for pointing this out. The geometric average, rather than the arithmetic mean, was proposed to be used in the work of Vandesompele et al. (2002), given its lesser susceptibility to extreme values. (Vandesompele, J., De Preter, K., Pattyn, F. et al. Accurate normalization of real-time quantitative RT-PCR data by geometric averaging of multiple internal control genes. Genome Biol 3, research0034.1 (2002). https://doi.org/10.1186/gb-2002-3-7-research0034)

Comments 5: One of most important questions is the quality of extracted RNA or DNA in cytological samples. The information on sample conservation is needed.

Response 5:

This technology has been extensively developed, and we published some of the first results in 2019 (Titov SE, Ivanov MK, Demenkov PS, et al. Combined quantitation of HMGA2 mRNA, microRNAs, and mitochondrial-DNA content enables the identification and typing of thyroid tumors in fine-needle aspiration smears. BMC Cancer. 2019;19(1):1010. Published 2019 Oct 28. doi:10.1186/s12885-019-6154-7). We believe that the quantity and quality of DNA and RNA are adequate for the specified analysis.

Comments 6: The conclusion is too general and not meaningful.

Response 5:

That is certainly the case. We have made some revisions to the conclusion.

Round 2

Reviewer 2 Report

Comments and Suggestions for Authors

The authors have considered the comments and modified the text accordingly.  I have no further comments.